# Microstructure Evolution and Mechanical Properties of Austenite Stainless Steel with Gradient Twinned Structure by Surface Mechanical Attrition Treatment

**DOI:** 10.3390/nano11061624

**Published:** 2021-06-21

**Authors:** Aiying Chen, Chen Wang, Jungan Jiang, Haihui Ruan, Jian Lu

**Affiliations:** 1School of Material Science & Engineering, University of Shanghai for Science and Technology, Shanghai 200093, China; wangchenakaai@163.com (C.W.); jiangjg01@163.com (J.J.); 2Department of Mechanical Engineering, The Hong Kong Polytechnic University, Hong Kong 810005, China; haihui.ruan@gmail.com; 3Department of Mechanical and Biomedical Engineering, City University of Hong Kong, Hong Kong 999077, China; jianlu@cityu.edu.hk

**Keywords:** austenitic stainless steel, surface mechanical attrition treatment, multi-scaled twin, mechanical property, strengthening

## Abstract

Gradient structures in engineering materials produce an impressive synergy of strength and plasticity, thereafter, have recently attracted extensive attention in the material families. Gradient structured stainless steels (SS) were prepared by surface mechanical attrition treatment (SMAT) with different impacting velocities. The microstructures of the treated samples are characterized by gradient twin fraction and phase constituents. Quantitative relations of gradient microstructure with impacting time and mechanical properties are analyzed according to the observations of SEM, TEM, XRD, and tests of mechanical property. The processed SSs exhibited to be simultaneously stiff, strong, and ductile, which can be attributed to the co-operation of the different spatial distributions of multi-scaled structures. The formation of gradient twinned structure is resolved and the strengthening by gradient structure is explored.

## 1. Introduction

Austenitic stainless steels (SS) have been widely developed in various engineering applications because of their excellent workability, corrosion resistance, superior low-temperature toughness, etc. [1,2]. However, the low strength of the austenitic SSs is a dominant drawback in the fields of commercial reactor, vessel internals, and coolant piping [3,4,5]. Conventionally, austenitic SSs cannot be hardened by heat treatment, but by cold working, which results in a significant increase of dislocation density, and strain-induced martensite transformation in some types of austenitic SSs [6,7]. The transformation strengthening combining with refinement of grain sizes can also be achieve by repeated hot-worked, cold-worked and low-temperature treatment [8]. However, these strengthening methods exhibit several disadvantages, such as low cold formability, deterioration of high-temperature stability, decrease of corrosion behavior, etc., [9,10]. Therefore, the high-performance SSs with new strengthening mechanisms need to be developed.

A recently emerged concept of coherent twin boundary (TB) is incorporated into the design for strengthening SSs [11,12]. TBs are special internal boundaries, in which strengthening is based on dislocation-TB interactions and the toughening is originated from the slips of partial dislocations along the coherent TBs [13,14,15]. Thereby, high-density twins with a finer spacing, especially down to the nanometer scale, can lead to a significant enhancement of flow strength and improvement of ductility. However, high density of refined twins in engineering materials is not easily formed [16,17,18]. At present, high-density nanotwins are obtained under the extreme conditions of the model materials, such as the copper foil with a thickness of several tens of micrometers [19,20]. In order to develop an advanced structural steel with high strength and large ductility, an austenite SS composed of multi-scaled twin bundles and nanocrystalline is produced in one-step by an advanced technology of surface mechanical attrition treatment (SMAT) with high-impacting frequency [21,22]. The SMAT set-up was first introduced by K. Lu and J. Lu in 1999, which was based on the vibration of spherical balls using high-power ultrasound [23]. The randomly flying balls were blasted on the surface propelling by an extremely high sound pressure as the driving force, then nanocrystalline (NC) layer was produced on a variety of metallic materials, together with a gradient grain size distribution [24,25,26]. In order to control and regulate the microstructure, a number of parameters are involved in SMAT, including ball size, impact velocity, ball density, impact kinetic energy, and so on [21,27]. Large number of research works show that the hardness and tensile strength of SMATed materials significantly increase, and strongly depend on the kinetic energy of balls and shots [21,24,25]. The high-frequency SMAT generates very high-impacting velocity and thereafter resulting in large local strain rate, which is the critical factor to induce twinned structure [21,27,28,29]. In this work, microstructures evolution and mechanical properties of the SMATed SS by different impacting velocities are studied. Quantitative relationships between impacting velocity with the grain size, martensite transformation, and twinned structure are provided. The strengthening of the gradient nanotwinned structure is proposed. These results are very important for better understanding of the deformation behavior of gradient twinned structure.

## 2. Materials and Methods

The chemical compositions of austenite 304 SS is 19.1Cr, 9.9Ni, 0.04C, 0.15Si, 1.40Mn, 0.005S, 0.026P and balance Fe (all in mass %). The 304 SS sheets (70 mm × 50 mm × 1 mm) were treated by SMAT with a high frequency of 20 kHz. The impact velocity was calculated about 7 m s^−1^ for austenite SS balls and 10 m s^−1^ for bearing steel balls [27]. The different impacting velocities originate from the different material properties of balls, where bearing steel is harder than austenitic stainless steel, thus, induces a larger impacting velocity. The 304 SS sheets were treated by different durations with a number of 100 balls on both sides of the 304 SS sheets. When treating the samples, the SMAT was carried out in a symmetrical time interval of 0.5 s from front to back sides, thereafter, a symmetrical microstructure is obtained. The detailed parameters are provided in Table 1. The residual strain, defined by
(1)$$\varepsilon =\frac{4}{3}\ln\left(\frac{\nu_{ο}}{\nu_{f}}\right)$$
was used to measure the macroscopic plastic strain of the materials after SMAT, where ν_0_ and ν_f_ are the initial and final volume of the treated samples, respectively.

The samples for the tensile tests were cut into dog-bone-shaped specimens with a gauge length of 30 mm and a cross section of 6 mm × 1 mm. Tensile tests were performed at room temperature at a strain rate of 6.7 × 10^−4^ s ^−1^ using a MTS Alliance RT/50 Materials Testing System (MTS, Eden Prairie, MN, USA). Four specimens were used to obtain consistent stress–strain curves. Vickers microhardness was measured by applying a load of 50 g for 15 s and taking the average of five separate measurements on each depth. A Philips Xpert X-ray diffractometer (XRD, Philips, Eindhoven, The Netherlands) with Cu K_α_ radiation was used to determine the phase constitution and estimate the phase content. Scanning electron microscopy (SEM) observations were performed with a HITACHI S-4200 field emission scanning electron microscope (Hitachi Ltd., Tokyo, Japan). Transmission electron microscopy (TEM) observations were made with a JEM 2010 transmission electron microscope (JEOL, Tokyo, Japan) with an operating voltage of 200 kV. The TEM foils were ion-thinned at low temperature. The statistical distribution of the grain size was estimated from TEM images, and twin fraction of the ultra-fine twins with spacing (λ_1µm_) lower than 1 µm was determined by calculating the area fraction of grains containing twins from SEM observations.

## 3. Results

### 3.1. Mechanical Properties

Figure 1 displays the hardness variation as a function of treatment time of SMATed 304 SS samples by SS balls. The hardness distributions of all the treated samples present concave-shaped curves. Three characteristics are pointed out. One is the enhanced hardness at center for all the treated samples, indicating a bulk strengthening. Second is the saturation hardness. Although the maximal hardness is high up to 476 HV at the surface for the SMATed specimen with 20 min, but the other specimens with 12 and 15 min have the very close value. Third is the gradient change of the hardness, where the hardness amplitude from the surface to center is obviously steeper with the increase of treating time.

The engineering stress–strain curves of the SMATed samples by SS balls (in Figure 2a) show a very obvious enhancement of the strength with treating time, where the yield stress (σ_0.2_) ranges from 540 to 910 MPa, much higher than that of the original 304 SS steel (267 MP). The ultimate tensile stress (σ_b_) increases from 684 MPa (0 min) to 810 MPa (1 min) and 980 MPa (20 min). Moreover, the total elongations to failure (ε_b_) of the SMATed specimens with treating times of 1–20 min also maintain high levels from 61% to 32%. The good combination of high strength and large ductility is owing to the higher strain-hardening capacity, as shown in the true stress–strain curves of Figure 2b. The strain-hardening exponent, n, can be estimated, according to the Ludwik–Hollomon equation,
(2)$$\sigma =\sigma_{Y}+{K\varepsilon }^{n}$$
where σ and ε are the true stress and strain, σ_Y_ is the initial stress, and K is the strength constant. As to the SMATed samples by SS balls, the n reaches the maximum of 0.78 for the one treated by 10 min, and then falls back to initial level of 0.73 for the one treated by 20 min. More interestingly, the true stress of all the SMATed samples reach a close saturation value of 1273 MPa, as illustrated in Figure 2b.

When using bearing steel balls with a higher impact velocity to treat the sample, a rapid strengthening is found, shown in Figure 3a. Specifically, the SMATed sample treated by 1 min exhibits a 701 MPa, while that is 540 MPa for the stainless steel balls. Compared with the SMATed sample treated by stainless steel balls with 10 min, the sample by bearing steel balls presents higher yield strength (950 MPa) and ultimate strength (1070 MPa), together with a close elongation to fracture (37%). Correspondingly, the strain-hardening exponent, n, also is larger than the samples with the same treating time as shown in Figure 3b. These results indicate that the strength increases while the elongation are not sacrificed. Figure 4a gives the change tendency of strength with treatment time, where the yield strength gradually increases and tends to a saturation value of 910 (stainless steel balls) and 950 MPa (bearing steel balls). These ultimate strengths are 980 (stainless steel balls) and 1070 MPa (bearing steel balls), respectively. Additionally, the strengths, including yield and ultimate strengths, both present a linear relationship with elongation to fracture within the treatment time of 15 min for stainless steel balls and 10 min for bearing steel balls, and then reach saturation, as seen from Figure 4b.

Figure 5 shows the fracture morphologies of the SMATed samples treated by stainless steel balls with 10 min, which exhibits a mix fracture, characterized by tearing lips, microcracks, and dimples. The facture surface is mostly composed of the ductility fracture characteristic after tensile test (in Figure 5a), but the dimples are of much smaller size, and exhibit a graded increase from surface to center. Tearing bands are found at the surface with a thickness of 5 µm, and then shallow and parabolic dimples are observed at the subsurface, as given in Figure 5b. At the subsurface of 150 µm from the surface, large microcracks are formed, as shown in Figure 5a, where deep shearing edges and fine shearing bands are clearly found, as shown in the magnification of Figure 5c. Some deep dimples with microcracks are observed in the subsurface and interior, as indicated by arrows in Figure 5c,d.

### 3.2. Microstructure

The mechanical behaviors of the SMATed samples are controlled by the microstructures. Figure 6a shows the XRD patterns of the as-received 304 SS and SMATed samples by stainless steel balls. The as-received 304 SS is primary composed of γ-austenite (fcc) phase, and a slightly α’-martensite phase. However, α’-martensite transformation occurs during SMAT, and the content of α’-martensite phase increases with the increase of treatment time until to a saturation level of 35%, as shown in Figure 6b. Similarly, the SMATed samples by bearing steel balls also exhibit the same tendency, while the contents of α’-martensite phase are relatively smaller than these SMATed samples by stainless steel balls.

The morphologies of the as-received 304 SS and SMATed samples by stainless steel balls are shown in Figure 7, which are selected from the distance of 50 µm close to the surface. The as-received 304 SS shows equiaxed grains with an average grain size of 50 μm (in Figure 7a). But many shear bands occur in the samples after SMAT, as indicated by arrows in Figure 7b. When the treatment time increases, denser and finer shear bands are observed, as shown in Figure 7c,d. Some shear bands cut through the whole grains in one parallel direction, and some intersect with each other in an angle of 70°, as displayed by solid arrows and dot line arrows in Figure 7c,d respectively. It should be noted that the multi-scaled shear bands are formed in the bulk material in the SMATed specimens, but the area fraction gradually decreases.

The detailed microstructures of the SMATed samples are analyzed by TEM observations, as given in Figure 8. Typical surface microstructure of the SMATed sample by stainless steel balls with treatment time of 10 min is composed of nanocrystalline (NC)/ultra-fined grains (UFG) with an average grain size of 180 nm, as shown in Figure 8a, and presents random crystallographic orientation, as indicated by the diffraction rings of the selected-area electron diffraction (SAED) pattern in Figure 8a. High density of dislocations is observed in the NC/UFG, and dislocation cells also appear, as indicated in Figure 8a. In the deeper zone, multi-scaled twins with single and intersect slip systems are observed. The typical morphologies of twins are given in Figure 8b,c, which are obtained from the zones at 50 µm and 300 µm depths from the surface, respectively. The deformed twins are characterized by finer single twins with higher density in the vicinity of surface, and gradually transit to coarser intersected twins in deeper zone. The dominant spacing (λ) of the single twin is lower than 100 nm (in Figure 8b). Intersected twins present an angle of about 70°, as shown in Figure 8c. High density dislocations are also observed at TBs and interior of twins, as marked by dot arrows in Figure 8c. These results suggest that the high-density shear bands observed in SEM images are the single or intersect twins.

## 4. Discussion

### 4.1. Effect of Impacting Velocity on Graded Structure

According to the results of hardness distribution, it can be reasonably thought that the microstructure of SMATed specimens exhibit a gradient change, involving NC/UFG grains at the surface, multi-scaled twins and dislocation substructures interior, as observed in Figure 8. The gradient structure is caused by the attenuated strain rate of SMAT [21,27]. During impacting deformation, three competitive mechanisms control the microstructure of the austenitic SS, basically, slip, twinning, and transformation [30,31], which are all relative to the impacting strain rate. Two impacting velocities are used in this experiment. Correspondingly, the higher impact velocity provides greater impacting force and larger strain rate, thereafter, increases the probability of deformation twins. Comparing the microstructure of the two types of samples by the low- and high-impacting velocities, the martensite transformation is inhibited in the higher-impacting velocity, exhibiting a relatively smaller content as shown in Figure 6b. The grain sizes at the surfaces of the two samples exhibit a gradual decrease with impacting time, and the high-impacting velocity results in a rapid refinement, as shown in Figure 9a. The grain sizes reach a saturation size of 138 nm by the high-impacting velocity with 10 min and 150 nm by the low-impacting velocity of 20 min, showing a close grain size. On the other hand, a large number of stacking faults and deformed twins are formed inside the austenite matrix, especially at the depth of 50–150 µm depth from the surface for all the SMATed samples, as shown in Figure 9b,c. The twin fraction increases significantly with increasing impacting velocity, especially in the center zone of the samples. These results suggest that the twinning deformation is sensitive to the impacting velocity.

Since the multi-scaled twins traverse the entire thickness of the sheets, the strain should also penetrate the bulk materials. The residual strains of the samples after SMAT are shown in Figure 10, where the plastic strain increases with treating time, and attains a saturation strain of 0.17 for the sample by stainless steel balls with 15 min and 0.20 for the one by bearing steel balls with 10 min. The occurrence of residual strain means that bulk volume is reduced after SMAT, and the higher impact velocity generates larger strain. The saturation strain implies that the impacting energy or impacting time cannot further generate plasticity of the sheet due to the strengthening of the treated materials, and thereafter, an upper limit of grain size and twin fraction is reached.

### 4.2. Synergetic Reinforcement of the Graded Structure

The superior mechanical properties can be attributed to the complex microstructure induced by the SMAT process, such as the gradient distribution of twin fraction (Figure 9b,c) and martensite phase (Figure 6b). The outer NC layer, especially the hard α′ phase, provides superior stiffness and hardness, and exhibits a limited plastic deformation by grain rotation and grain boundary slide [28]. The high-density twins lead to a significant enhancement in flow strength and an improvement of ductility by the dislocation accumulation at TBs [18,24]. The junctions between the various structures are clearly “functionally graded,” that is, they possess a gradual spatial change in properties, which can promote an effective transitional region for stress redistribution [32,33,34]. The hardness variation of SMATed sample by stainless steel with 10 min exhibits obviously gradient change during different stages of tensile deformation, indicating an important smoothing role of the gradient structure in deformation, as shown in Figure 11. Notably, the NC/UFG surface also exhibits an increased hardness during the tensile deformation, which means the strengthening still operates. During the deformation process, the multiple-scale twins strengthen by secondary twinning and intersected twinning, and change the orientation of the subgrains to transfer the deformation to the adjacent grains or subcrystals [31,33,34]. Through relieving the stress concentration by the graded structure, the extension of deformation is achieved, which endows the SMATed samples high plasticity (Figure 2a and Figure 3a). More importantly, the refinement by the twin boundaries, such as the intersected twinning or multiple twinning, can significantly enhance the hardness and strength by the dislocation pile-up at TBs, similar to the Hall-Petch rule [18]. Correspondingly, the hardness gradually increases during tensile deformation, until their hardness is close to the surface NC/UFG layer, as observed in Figure 11. Hence, graded distribution of multi-scaled twins is necessary to obtain both high strength and high plasticity by multiple deformation behaviors to release stress concentration compared to the uniform material.

## 5. Conclusions

Effect of different impacting velocities during SMAT on microstructure and mechanical properties of 304 stainless steels were investigated. The following conclusions can be obtained.
(1)The microstructures of SMATed sample are composed of NC/UFG surface and multi-scaled twins interior in a gradient distribution, characterized by graded twin fraction and martensite phase. The high-impacting velocity stimulates the formation of ultra-fined twins and inhibits the martensite transformation, where the twin fraction is high up to 72% for the sample by bearing steel balls with 10 min.(2)The tensile strength of the SMATed sample increases with the treatment time, and reaches a saturation level. The high-impacting velocity results in a simultaneous increase of strength and ductility compared with the low-impacting velocity.(3)The impressive combination of strength and ductility originates from the juxtaposition of multiple reinforcing microstructure. The gradient structures of twin fraction result in a graded change of hardness, playing an important smoothing role during tensile deformation.

## Figures and Tables

**Figure 1 nanomaterials-11-01624-f001:**
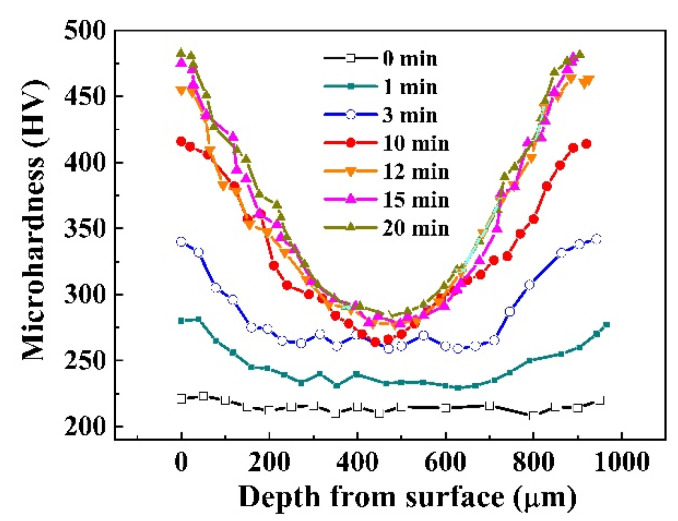
Dependence of Vichers microhardness distribution with depth of SMATed stainless steels by SS balls.

**Figure 2 nanomaterials-11-01624-f002:**
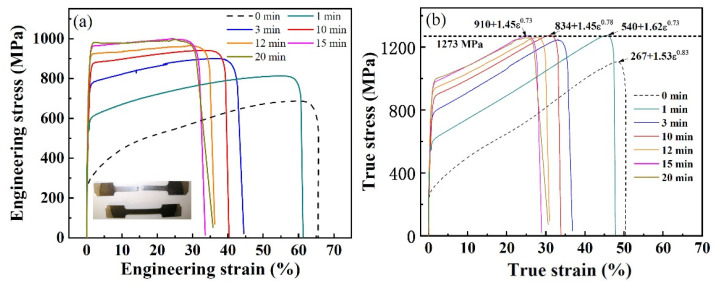
Tensile curves of 304 SS before and after SMAT by SS balls. (**a**) Engineering stress–strain curves; (**b**) true stress–strain curves; the insets in (**a**) are the SMATed samples before and after the tensile test.

**Figure 3 nanomaterials-11-01624-f003:**
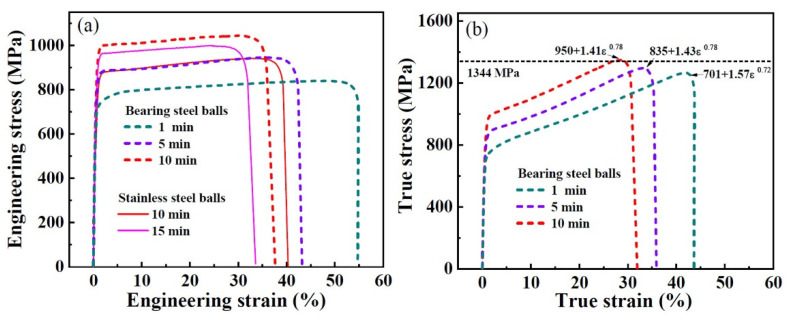
Tensile curves of SMATed 304 SS by bearing steel balls. (**a**) Engineering stress–strain curves; (**b**) true stress–strain curves.

**Figure 4 nanomaterials-11-01624-f004:**
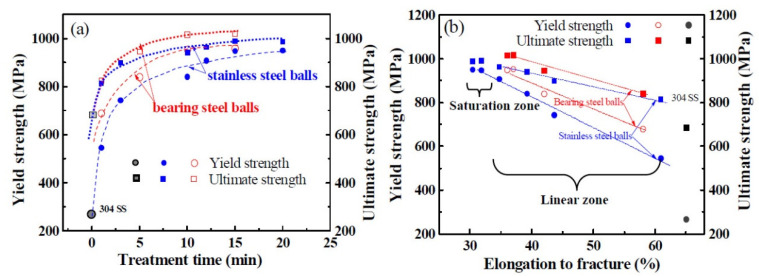
Relationship of strength with treatment time (**a**) and elongation to fracture (**b**).

**Figure 5 nanomaterials-11-01624-f005:**
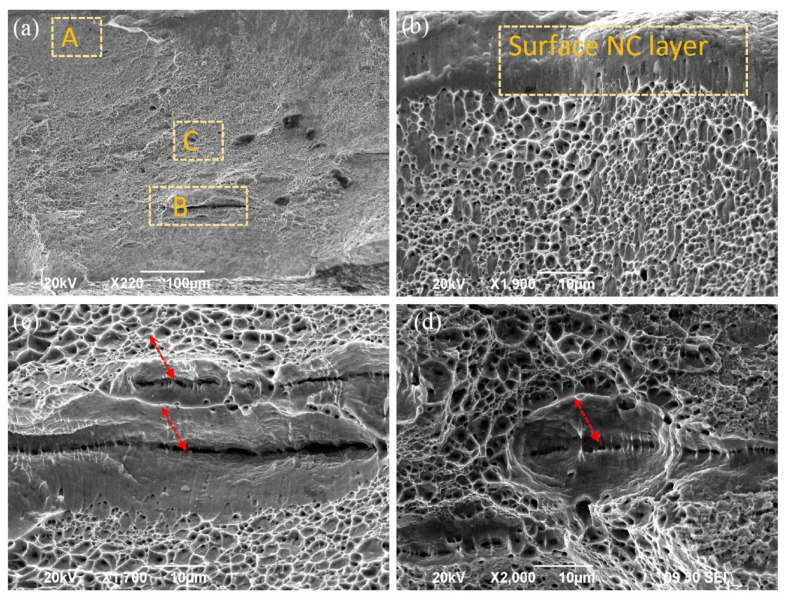
SEM observations of fracture morphology of SMATed sample by stainless steel ball with 10 min. (**a**) Global morphology; (**b**–**d**) magnifications of the corresponding A, B, C zones in (**a**), showing the morphologies of surface NC layer, subsurface, and center zones.

**Figure 6 nanomaterials-11-01624-f006:**
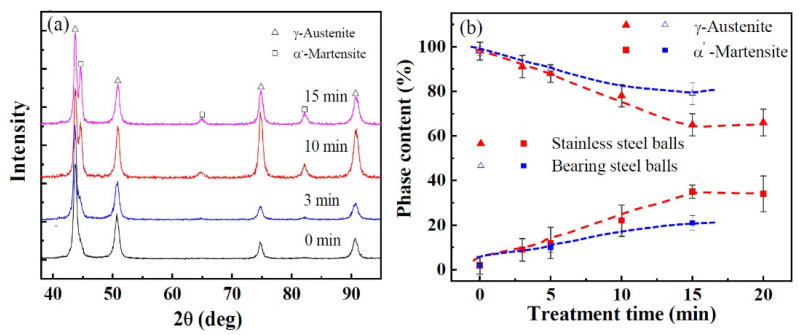
XRD patterns (**a**) and phase content (**b**) at the surfaces of the SMATed samples by stainless steel balls.

**Figure 7 nanomaterials-11-01624-f007:**
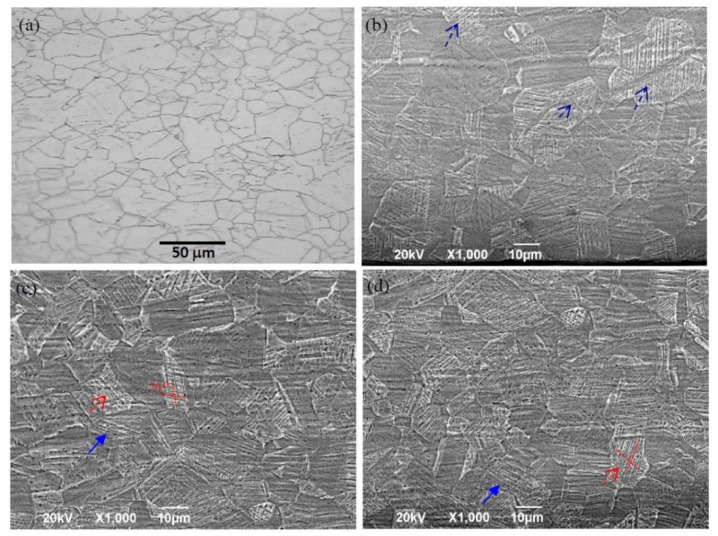
Morphologies of SMATed samples by stainless steel balls from the distance of 50 µm to the surface. (**a**) 0 min; (**b**) 3 min; (**c**) 10 min; (**d**) 15 min.

**Figure 8 nanomaterials-11-01624-f008:**
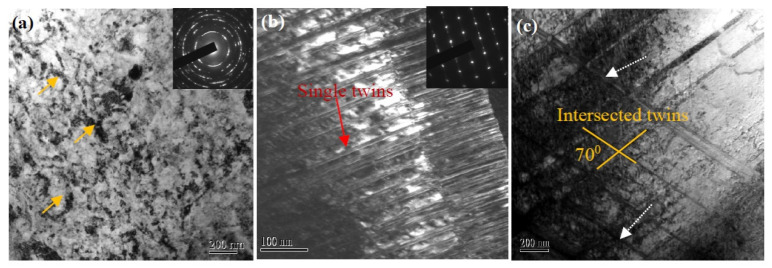
Typical microstructures of the SMATed samples by stainless steel balls with 10 min. (**a**–**c**) Bright-field TEM images at depths of 0, 50, and 300 µm, respectively.

**Figure 9 nanomaterials-11-01624-f009:**
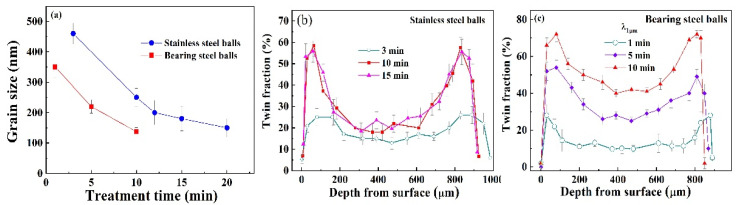
Grain size distribution at the surface (**a**), twin fraction with λ_1__µm_ lower than 1 µm of the SMATed samples by stainless steel balls (**b**) and bearing steel balls (**c**).

**Figure 10 nanomaterials-11-01624-f010:**
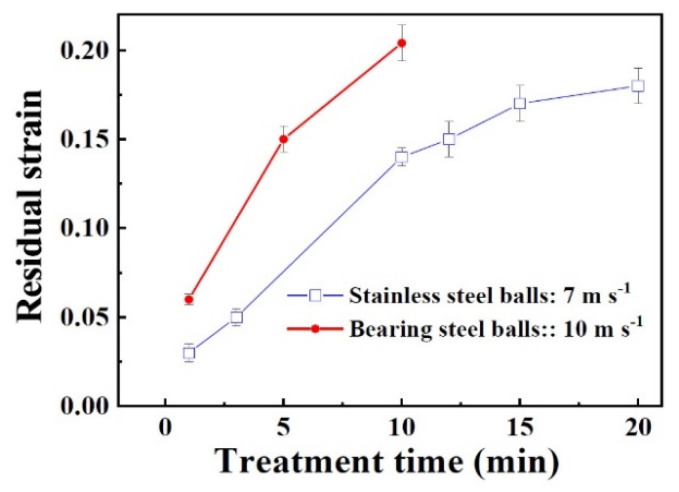
Dependence of residual strain of SMATed samples with treatment time.

**Figure 11 nanomaterials-11-01624-f011:**
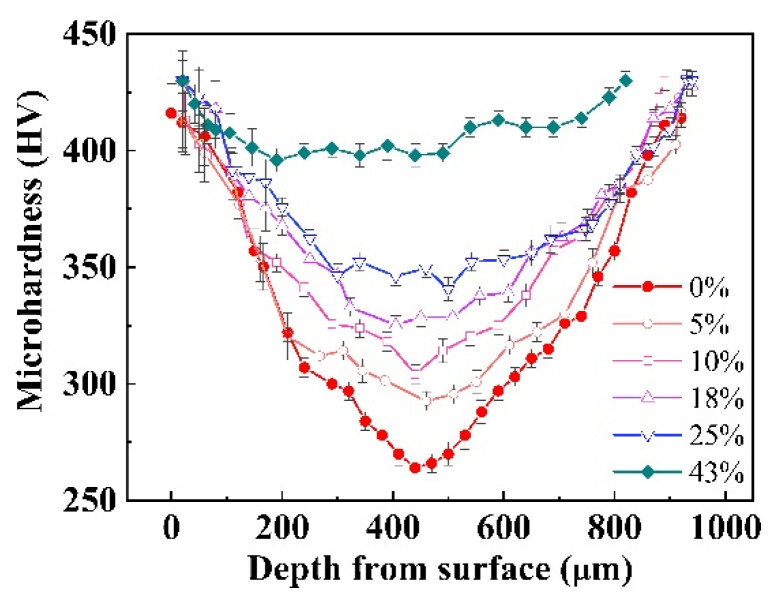
Hardness distribution with the dependence of tensile strain of the SMATed sample by stainless steel balls with 10 min.

**Table 1 nanomaterials-11-01624-t001:** Processing parameters of SMAT.

Vibrating Frequency (kHz)	Impact Velocity (m s^−1^)	Ball Material	Diameter of Ball (mm)	Number of Ball (pcs)	Time Interval (s)	Treated Side
20	7	Stainless steel	3	100	0.5	Front and back sides
20	10	Bearing steel	3	100	0.5	Front and back sides

## Data Availability

The data presented in this study are available on reasonable requests from the corresponding author.

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
