# Peer review of "Microstructure Evolution and Mechanical Properties of Austenite Stainless Steel with Gradient Twinned Structure by Surface Mechanical Attrition Treatment"

_nanomaterials, 2021, doi:10.3390/nano11061624_

Round 1

Reviewer 1 Report

The paper deals microstructure evolution in stainless steels with gradient twinned structure by surface mechanical attrition treatment.

The topic is quite interesting and suitable for the journal.

I think the paper should be accepted after some minor modification in the introduction section

  1. Line 33: add a more general reference about stainless steel. I suggest: Di Schino, A. Manufacturing and application of stainless steels. Metals 2020, 10, 327
  2. Line 36: This sentence is not true at all: deformation induced martensite formation is not found in ALL austenitic grades. Please discuss about. Cold deformation process results in increase of dislocation density as a primary consequence.
  3. Line 40: many methods have been developed based on deformation induced martensite: in particular low temperature heat treatment results in grain refining. I suggest to add re following reference: Järvenpää, A.; Jaskari, M.; Kisko, A.; Karjalainen, P., Process and properties of reversion-treated austenitic stainless steels. Metals, 2020, 281.

Reviewer 2 Report

This paper presents the microstructure and mechanical properties of austenite stainless steel with gradient twin structure and contains some interesting results, but the following points should be reconsidered.

1) The authors used two types of balls in this study. I think that the authors wanted to investigate the effect of impact velocity on microstructure and mechanical properties. However, does the material properties of each ball have no effect?

2) The x-axis is “distance from surface” in Fig. 1, whereas that is “depth from surface” in Figs. 9 and 11. Which is correct?

3) The authors described that the overall microstructure of SMATed specimens is symmetrical. However, this is estimated based on the data of hardness and twin density, and the relationship between microstructure and depth from surface should be shown as a direct evidence.

4) The unit of twin density is expressed as “%” in this paper (Fig. 9). Is this correct? If it is twin “density”, the unit is “/m2”, and if the unit is “%”, it is twin “fraction”.

5) It is strange that the description of Fig. 10 (Page 8, Line 215) appears before the description of Fig. 9 (Page 9, Line 233). In addition, I cannot find the description of Fig. 9(a).

6) Page 9, Line 239. The authors described that the grain size shows less dependence to the impacting velocity. However, the grain size may depend on the impacting velocity based on the data shown in Fig. 9(a).

Round 2

Reviewer 2 Report

The revision is satisfying.